# ATP binding cassette transporters and uridine diphosphate glycosyltransferases are ancient protein families that evolved roles in herbicide resistance through exaptation

Samuel Caygill[1,2], Liam Dolan[1,2]*

**1** Gregor Mendel Institute, Vienna, Austria, **2** Department of Biology, University of Oxford, Oxford, United Kingdom

* liam.dolan@gmi.oeaw.ac.at

## Abstract

ATP-binding cassette (ABC) transporters actively transport various substances across membranes, while uridine diphosphate (UDP) glycosyltransferases (UGTs) are proteins that catalyse the chemical modification of various organic compounds. Both of these protein superfamilies have been associated with conferring herbicide resistance in weeds. Little is known about the evolutionary history of these protein families in the Archaeplastida. To infer the evolutionary histories of these protein superfamilies, we compared protein sequences collected from 10 species which represent distinct lineages of the Archaeplastida–the lineage including glaucophyte algae, rhodophyte algae, chlorophyte algae and the streptophytes–and generated phylogenetic trees. We show that ABC transporters were present in the last common ancestor of the Archaeplastida which lived 1.6 billion years ago, and the major clades identified in extant plants were already present then. Conversely, we only identified UGTs in members of the streptophyte lineage, which suggests a loss of these proteins in earlier diverging Archaeplastida lineages or arrival of UGTs into a common ancestor of the streptophyte lineage through horizontal gene transfer from a non-Archaeplastida eukaryote lineage. We found that within the streptophyte lineage, most diversification of the UGT protein family occurred in the vascular lineage, with 17 of the 20 clades identified in extant plants present only in vascular plants. Based on our findings, we conclude that ABC transporters and UGTs are ancient protein families which diversified during Archaeplastida evolution, which may have evolved for developmental functions as plants began to occupy new environmental niches and are now being selected to confer resistance to a diverse range of herbicides in weeds.

## Introduction

Populations of weeds are continually evolving resistance to herbicides [1]. Herbicide resistance mechanisms can be classified into 2 general groups: target site resistance (TSR) and non-target

**Data Availability Statement:** All relevant data are within the paper and its Supporting Information files.

**Funding:** This research was supported by a European Research Council (ERC) Advanced Grant De Novo-P (project number 787613) to LD from the European Commission. SC was supported by a British Biological Sciences Research Council (BBSRC) Scholarship through a doctoral training partnership (BB/M011224/1). The funders had no role in study design, data collection and analysis, decision to publish, or preparation of the manuscript.

**Competing interests:** L. D. is a co-founder of moa Technology Ltd. This does not alter our adherence to PLOS ONE policies on sharing data and materials.

site resistance (NTSR). Target site resistance (TSR) occurs as a result of mutations in the gene coding for the herbicide's target protein, which causes reduced inhibition of the herbicide target. Non-target site resistance (NTSR) can occur due to a variety of mechanisms, but ultimately results in a reduction in the amount of herbicide reaching its target protein, or by reducing damage caused by the herbicide to the plant [2]. Some of the NTSR mechanisms reported include reduction in the uptake or translocation of the herbicide, metabolic detoxification of the herbicide and sequestration of the herbicide to a location where it is not active, such as the vacuole [3–6]. Diverse genetic changes affecting NTSR genes can lead to their encoded proteins contributing to herbicide resistance. For example, mutations leading to gene overexpression, the expression of a hyperactive form of the protein, loss-of-function mutations or mutations leading to changes in substrate specificity, can all result in NTSR [4, 7, 8]. The genetic basis of NTSR is poorly understood, but changes in the expression of both ATP-binding cassette (ABC) transporters and uridine diphosphate (UDP) glycosyltransferases (UGTs) have been shown to confer resistance to herbicides in weeds [3–5].

ABC transporters are one of the largest protein families in plants and are conserved across all domains of life [9]. ABC transporter proteins use the energy of ATP hydrolysis to drive active transport of substances across membranes [10, 11]. In plants, ABC transporters transport nutrients such as sugars, amino acids and metal ions; secondary metabolites such as flavonoids, terpenoids and alkaloids; phytohormones such as auxins, cytokinins and gibberellins; proteins and peptides; and xenobiotics [12–14]. Plant ABC transporters are also involved in the translocation of toxic compounds, such as herbicides, into the vacuole and extracellular spaces, where they are no longer toxic [15, 16]. Numerous ABC transporters have been identified to compartmentalise herbicides in this way, as part of a detoxification process that has evolved in some weeds with NTSR to at least one herbicide.

Most ABC transporters are membrane bound and consist of four subunits–two transmembrane domains (TMDs) and two nucleotide-binding domains (NBDs) (S1 Fig) [17]. Generally, the substrate binds to the TMDs of the transporter on one side of the membrane. Binding of ATP to the NBDs then causes a conformational change in the transporter, closing the TMDs around the substrate. Subsequently, the hydrolysis of ATP causes another conformational change in the transporter that opens up the TMDs on the opposite side of the membrane, releasing the substrate [10, 18]. It is with this mechanism that ABC transporters are able to transport herbicides as part of a herbicide-resistance-conferring detoxification process.

Agricultural fields have been treated with herbicides since the 1940s, exposing plants to chemical diversity that they would not have previously encountered. Some ABC transporters catalyse the transport of a wide variety of synthetic compounds across membranes that they would not have encountered in their evolutionary history. ABC transporters are one of the most diverse protein families in plants, exhibiting significant variation in sequence, structure, and function. In plants, ABC transporters have been classified into 8 subfamilies (ABCA, ABCB, ABCC, ABCD, ABCE, ABCF, ABCG and ABCI) [10, 11, 19, 20]. Previous phylogenetic analyses in plants have shown all ABC transporter subfamilies, excluding ABCI, constitute monophyletic clades [10, 11, 19]. However, previous phylogenetic analyses of these protein families focused primarily on angiosperms. To gain a more comprehensive understanding of the evolution of this protein family, here we carry out a phylogenetic analysis of ABC transporters from across the Archaeplastida.

Uridine diphosphate (UDP) glycosyltransferases (UGTs) are part of the largest family (Family 1) of the glycosyltransferase superfamily, and this subset of UGTs have been widely associated with herbicide resistance [21, 22]. UGTs are present in bacteria, fungi, plants, and animals and catalyse the transfer of glycosyl residues from activated nucleotide sugars, such as uridine diphosphate glucose, to acceptor molecules [23]. Acceptor molecules utilised by UGTs

include flavonoids, phenolic acids, plant hormones such as IAA, xenobiotics, and other small molecules such as alkaloids, terpenoids and cytokinins [24]. While UGTs often transfer glycosyl residues to various endogenous compounds, a variety of exogenous xenobiotics, including herbicides, can also act as the acceptor molecules of UGTs. The addition of a glycosyl residue to a herbicide has the general effect of making it more hydrophilic and less toxic. Therefore, mutations in genes coding for UGTs that enable them to glycosylate herbicides, or mutations that cause overexpression of UGTs that can already glycosylate herbicides, can contribute to NTSR in weeds.

UGTs are classified by the presence of a C-terminal consensus sequence, the UDP-glycosyltransferase signature, known as the 'PSPG' motif (Plant Secondary Product Glycosyltransferase). The PSPG motif is made up of 44-amino acids and is thought to be the site of binding of the UDP moiety of the sugar molecule to the enzymes [25–27]. On the other hand, the N-terminal regions of UGTs is the domain which is involved in the recognition and binding of a diverse array of acceptor molecules. Therefore, the N-terminal domain is generally more variable than the C-terminus [28]. UGTs catalyse O-, S- and N-linkages and can also form different linkages with the same acceptor molecule [29]. For example, *A. thaliana* UGT72B1 has been reported to form an N-glucosidic bond with 3, 4-dichloroaniline and an O-glucosidic bond with 3,4-dihydroxybenzoic acid [30]. Previous phylogenetic analyses of plant UGTs has found this protein family to diverge into at least 20 distinct monophyletic groups [25, 26]. However, the taxon sampling of these studies has been limited to angiosperms, and so conclusions cannot be made regarding the evolutionary history of this protein family in the Archaeplastida lineage.

Here we report the phylogenetic relationships, of both the ATP-binding cassette transporter and uridine diphosphate glycosyltransferase protein superfamilies within the Archaeplastida lineage. We discovered that the current clade diversity of the ABC transporter superfamily existed in the last common ancestor of the Archaeplastida. This indicates that the ABC transporters diversified into the extant sub-families before the evolution of the last common ancestor of the Archaeplastida, extant 1.6 billion years ago [31–33]. By contrast UGT proteins are only present in the streptophyte lineage among the Archaeplastida and diversified in the vascular plant lineages. On the other hand, we show that UGTs are present in the streptophyte lineage of the Archaeplastida but absent from non-streptophyte lineages. This suggests either that this protein family, which exists in other eukaryotic lineages, was lost from the glaucophyte, rhodophyte and chlorophyte lineages, or that these proteins originated through horizontal gene transfer into the last common ancestor of the streptophytes, extant approximately 1.1–1.2 billion years ago in the Proterozoic era [34–39]. Furthermore, we show that the UGT family subsequently diversified throughout streptophyte evolution, mostly in the vascular lineage. This analysis suggests that gene superfamilies which evolved in the Proterozoic era, are now undergoing natural selection caused by herbicide application which is leading to the evolution of herbicide resistant weeds in agricultural systems.

## Materials and methods

### Data resources

Amino acid sequences of proteins from *A. thaliana* were collected from TAIR10 [40] (https://www.arabidopsis.org/). Sequences from *O. sativa* were retrieved from the rice genome annotation project [41]. Sequences from the liverwort *M. polymorpha* were collected from MarpolBase (http://marchantia.info/). Sequences from the hornwort *Anthoceros agrestis* were collected from [42] (https://www.hornworts.uzh.ch/en/download.html). Sequences from the streptophyte alga *Klebsormidium nitens* were obtained from the *K. nitens* genpme webpage

[43]. (http://www.plantmorphogenesis.bio.titech.ac.jp/~algae_genome_project/klebsormidium/). Sequences from the lycophyte *Selaginella moellendorffii*, the moss *Physcomitrium patens* and the chlorophyte alga *Chlamydomona reinhardtii* were retrieved from Phytozome 12 [44] (https://phytozome.jgi.doe.gov/pz/portal.html). Protein sequences from the rhodophyte alga *Cyanidioschyzon merolae* were collected from the *C. merolae* genome webpage [45] (https://www.genome.jp/kegg-bin/show_organism?org=cme). Sequences from the glaucophyte alga *Cyanophora paradoxa* were collected from PhycoCosm [46] (https://phycocosm.jgi.doe.gov/Cyapar1/Cyapar1.home.html).

## Sequence collection

Family 1 UGT and ABC transporter protein sequences from *A. thaliana* and *O. sativa* [40, 41], were used to perform BLASTP searches against the predicted proteomes of *S. moellendorffii*, *M. polymorpha*, *A. agrestis*, *P. patens*, *K. nitens*, *C. reinhardtii* and *C. merolae*. A minimum E-value threshold of $1e^{-10}$ was employed. The initial list of sequences for each species collected in this way was then used to 'self-blast' the predicted proteomes of that species in order to retrieve sequences belonging to species-specific Family 1 UGT and ABC transporter sub-families. Using InterProScan 92.0 [47], each Family 1 UGT sequence was checked for the UDP-glycosyltransferase domain (SSF53756), while each ABC transporter was checked for the presence of the ABC transporter pfam domain (PF12848). All sequences that did not contain the searched-for domains were removed before further analysis.

## Sequence alignment

Collected amino acid sequences were aligned using the programme MAFFT [48] and its FFT-NS-2 algorithm. Alignments were then visualised using Unipro ugene v. 45.1 [49]. Sequences which could then be seen to be lacking important functional residues were then removed. Sequences were trimmed manually, with large non-conserved regions being removed, leaving only mostly conserved regions without gaps.

## Phylogenetic analysis

The final trimmed amino acid sequence alignments were subsequently used in a maximum-likelihood phylogenetic analysis using the software PHyML 3.0 [50]. The parameters used were an estimated gamma distribution, the LG+G+F model of amino acid substitution, and a $Chi^2$-based approximate likelihood ratio test (aLRT). The resulting phylogenies were visualised in FigTree v1.4.4 [51] and annotated in Inkscape v1.0.2 [52].

## Results

### 960 ABC transporter and 552 UGT sequences were identified in the genomes of 10 species of Archaeplastida

We first set out to investigate the evolution of ATP-binding cassette (ABC) transporter and uridine diphosphate (UDP) glycosltransferase (UGT) gene families in the Archaeplastida lineage. The Archaeplastida is the lineage which includes glaucophyte algae, rhodophyte algae, chlorophyte algae and the streptophytes. ABC-transporter protein-encoding sequences were identified from 10 selected species (Table 1), representing distinct Archaeplastida lineages. UGT sequences were identified in the genomes of 6 of these 10 Archaeplastda species. There were 60 ABCs and 3 UGTs in the genome of the streptophyte alga *Klebsormidium nitens*; 100 ABCs and 50 UGTs in the liverwort *Marchantia polymorpha*; 85 ABCs and 15 UGTs in the moss *Physcomitrium patens*; 166 ABCs and 163 UGTs in the lycophyte *Selaginella*

**Table 1. List of members of Archaeplastida used in the analysis.**

| Species | Classification | Genome (Mb) | Protein-coding genes | ABCs | ABCs (% PCG) | UGTs | UGTs (% PCG) | References |
|---|---|---|---|---|---|---|---|---|
| *Arabidopsis thaliana* | Angiosperm eudicot | 135 | 25,498 | 120 | 0.471 | 121 | 0.475 | [53] |
| *Oryza sativa* | Angiosperm monocot | 321 | 35,681 | 124 | 0.339 | 200 | 0.561 | [54] |
| *Selaginella moellendorffii* | Lycophyte | 212.6 | 22,285 | 166 | 0.745 | 163 | 0.731 | [55] |
| *Anthoceros agrestis* | Hornwort | 133 | 24,700 | 130 | 0.526 | 0 | 0 | [42] |
| *Physcomitrella patens* | Moss | 480 | 35,938 | 85 | 0.237 | 15 | 0.042 | [56] |
| *Marchantia polymorpha* | Liverwort | 225.8 | 19,138 | 100 | 0.523 | 50 | 0.261 | [57] |
| *Klebsormidium nitens* | Streptophyte alga | 117.1 | 16,215 | 60 | 0.370 | 3 | 0.0185 | [43] |
| *Chlamydomonas reinhardtii* | Chlorophyte alga | 120 | 15,143 | 47 | 0.310 | 0 | 0 | [58] |
| *Cyanidioschyzon merolae* | Rhodophyte alga | 16.5 | 5,331 | 17 | 0.319 | 0 | 0 | [45] |
| *Cyanophora paradoxa* | Glaucophyte alga | 98.4 | 25,518 | 111 | 0.435 | 0 | 0 | [46] |

Including their phylum (Classification), genome size (Genome), total number of protein-coding genes (Protein-coding genes), total number of ABC transporter proteins (ABCs), ABC transporter proteins as a percentage of total protein coding genes (ABCs % PCG), total number of UDP-glycosyltransferase proteins (UGTs), UDP-glycosyltransferase proteins as a percentage of total protein coding genes (UGTs % PCG) and the reference for each genome sequence.

*moellendorffii*; 124 ABCs and 200 UGTs in the monocot *Oryza sativa*; and 120 ABCs and 121 UGTs in the eudicot *Arabidopsis thaliana*. UGT sequences were not recovered from the genomes of the glaucophyte algae *Cyanophora paradoxa*, the rhodophyte alga *Cyanidioschyzon merolae*, the chlorophyte alga *Chlamydomonas reinhardtii*, or the hornwort *Anthoceros agrestis*. While no UGT encoding genes were identified in these species, 111, 17, 47, and 130 ABC transporter coding sequences were identified in these species respectively. Taken together these data indicate that the ABC transporters were present in the genome of the last common ancestor of the Archaeplastida, while UGT encoding genes have been present in the Archaeplastida lineage since the divergence of the streptophytes from the last common ancestor shared with the chlorophytes.

## Plant ABC transporters are ancient, and all 12 groups existed in the last common ancestor of the Archaeplastida

To investigate the evolutionary history of the ABC transporter protein family, we generated a multiple sequence alignment of 960 ABC transporter protein sequences, which we used to construct a Maximum Likelihood tree (Fig 1). Alignments were trimmed manually to remove large gaps and only a nucleotide-binding domain (NBD) of each sequence was retained.

The topology of the phylogenetic tree shows that plant ABC transporters diverged into 6 previously identified, monophyletic clades. Each of these monophyletic clades has been classified by a single sub-family of ABC-transporters, with each subfamily designated a letter. For example, the monophyletic clade comprising of ABCA proteins is designated clade A. However, ABC transporter sub-families ABCE and ABCF are sister clades, and together form a single monophyletic clade, designated clade E/F. Our analysis revealed the previously defined monophyletic clades A, B, C, D, E/F, G (Fig 1). We also identified 6 previously unreported monophyletic clades that comprise sequences from a single species (*C. paradoxa*–Cp) or two species (*C. paradoxa* and *C. merolae*–CpCm)–Cp1, Cp2, Cp3, Cp4, Cp5, and CpCm1 (Fig 1). ABCI proteins are the only previously reported sub-family that do not form a monophyletic clade. We found ABCI proteins to comprise two distantly related monophyletic groups (and therefore constitute a polyphyletic group) (Fig 1). Sequences from the 4 clades B, C, E/F and G are encoded in the genomes of all 10 species of Archaeplastida used in this analysis. This suggests that these four clades were present in the last common ancestor of the Archaeplastida (Fig 2).

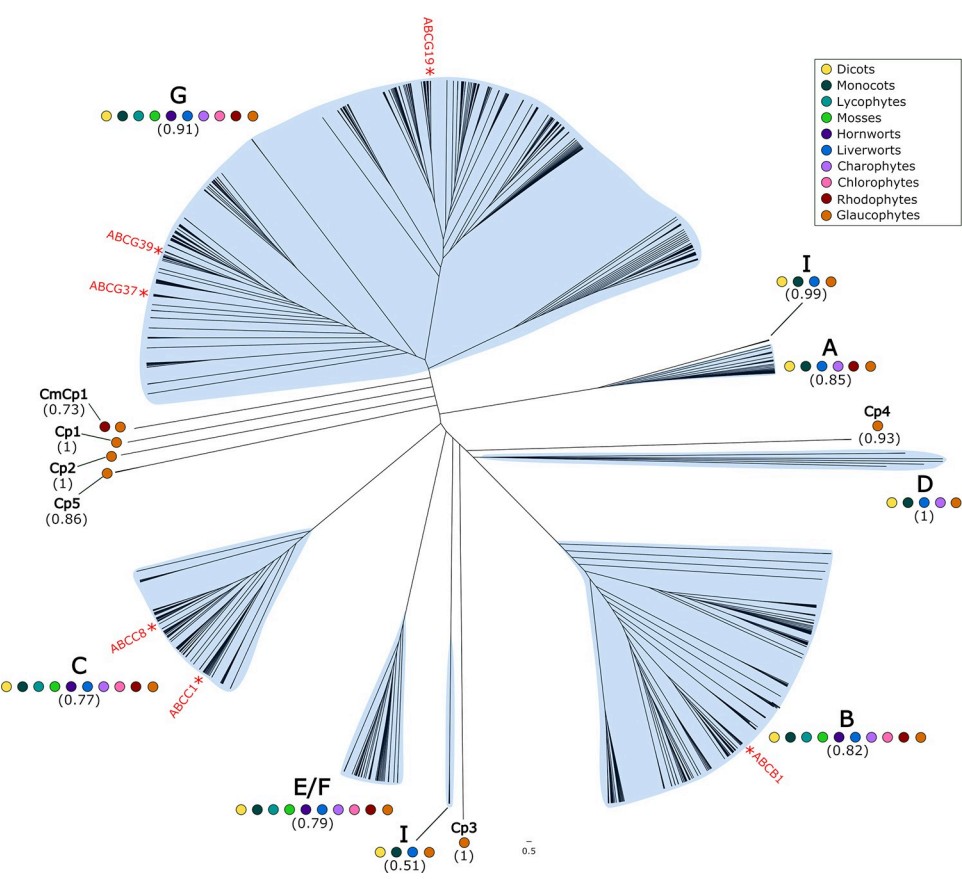

**Fig 1. Phylogenetic analysis of ABC transporter protein sequences in the Archaeplastida.** Unrooted cladogram of a maximum likelihood (ML) analysis of Archaeplastida ABC transporter proteins conducted by PhyML 3.0 [50] using an estimated gamma distribution parameter, the LG_G+F model of amino acid substitution, and a Chi$^2$-based approximate likelihood ratio (aLRT) test. Support values from the aLRT test for each clade are presented in parentheses. Protein sequences were aligned using MAFFT with the L-INS-I algorithm. ABC transporter classes are indicated by blue highlighting and letters. Coloured dots indicate the presence of sequences from different species in each clade. *A. thaliana* (yellow); *O. sativa* (dark green); *S. moellendorffii* (light blue); *P. patens* (light green); *A. agrestis* (dark purple); *M. polymorpha* (blue); *K. nitens* (light purple); *C. reinhardtii* (pink); *C. merolae* (maroon); *C. paradoxa* (orange). Confirmed NTSR genes or their *A. thaliana* homolog reported in Table 3 are indicated.

The presence of an ABCA protein sequences in the rhodophyte *C. merolae* and the glauco-phyte *C. paradoxa* suggests clade A is ancient and was present in the last common ancestor of the Archaeplastida. However, ABCA proteins are missing from the genomes of 4 species–*C. reinhardtii*, *A. agrestis*, *P. patens* and *S. moellendorffii*. The absence of ABCA proteins from these lineages likely represent independent gene loss events, following the divergence of these lineages from other land plants. There are no ABCA proteins encoded in the genome of the chlorophyte alga *C. reinhardtii*, but ABCA proteins are encoded by the streptophyte alga *K. nitens*. This suggests a loss of these proteins in the chlorophyte algae lineage sometime after its divergence from the last common ancestor of the chlorophytes and streptophytes. ABCA proteins are present in the genome of the liverwort *M. polymorpha* but absent from the other two bryophyte species included in the analysis–the hornwort *A. agrestis* and the moss *P. patens*. If hornworts are sister to the mosses and liverworts [42, 59], this topology suggests ABCA proteins were lost from the hornwort lineage following its divergence from the moss and liverwort lineages. ABCA proteins were then likely subsequently lost in the moss lineage following its divergence from the liverworts. ABCA proteins were found to be absent from the genome of *S. moellendorffii* but present in the other 2 vascular

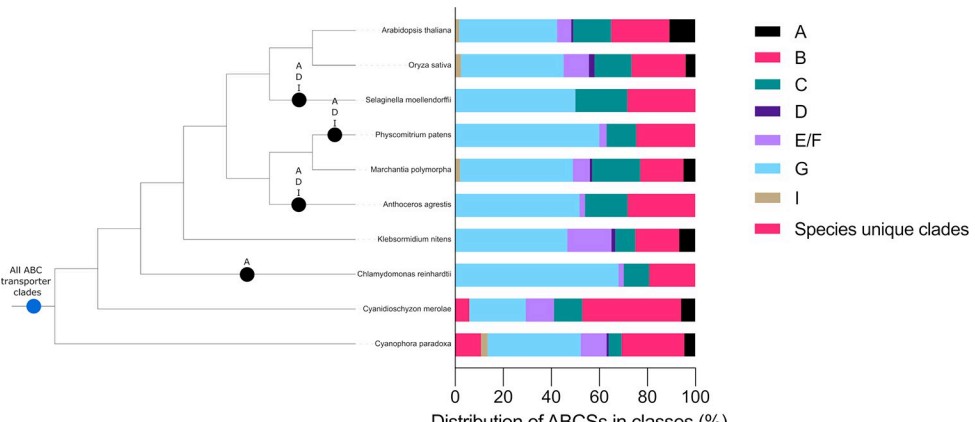

**Fig 2. ABC transporters clades were present in the last common ancestor of the Archaeplastida.** Cladogram of Archaeplastida showing ABC transporter clade origins and losses during plant evolution. Blue circles represent first appearances of a clade, black circles represent the loss of a clade in a particular lineage. Also shown are the relative distributions of ABC transporters into their respective clades for each species included in our analysis, where each distinct coloured block represents a separate ABC transporter protein clade.

plants included in the analysis–*A. thaliana* and *O. sativa*. This indicates a loss of ABCA proteins in the lycophyte lineage following its divergence from the angiosperm lineage (Fig 2). The absence of ABCA proteins in some lineages within the Archaeplastida is likely due to ABCA proteins being secondarily lost in these lineages, and hence ABCA proteins were likely already present in the last common ancestor of the Archaeplastida.

ABCD proteins were present in 5 lineages of the Archaeplastida, the dicots, monocots, liverworts, streptophyte algae and glaucophyte algae. The presence of ABCD proteins in the genome of the glaucophyte *C. paradoxa* suggest that ABCD proteins had already diverged from other ABC transporters in the last common ancestor of the Archaeplastida. However, ABCD sequences were not found in *C. merolae* or *C. reinhardtii*, suggesting loss of the ABCD subfamily in the rhodophyte and chlorophyte algal lineages. ABCD proteins were missing from the genomes of *A. agrestis*, *P. patens* and *S. moellendorffii*, as observed for ABCA proteins. Their absence from these lineages suggests recent gene losses in the hornworts, mosses, and lycophytes respectively. Angiosperm genomes were found to encode ABCD proteins, however, the other vascular plant included in the analysis, *S. moellendorffii*, does not encode any ABCD proteins, suggesting that this clade was lost in this lineage following the divergence of the lycophyte lineage from the other vascular plants (Fig 2). ABCD proteins were therefore likely present in the last common ancestor of the Archaeplastida lineage, but were subsequently lost in the rhodophyte, chlorophyte, hornwort, moss and lycophyte lineages.

Many putative ABCI encoding genes were identified in the genomes of the plant species in this analysis. However, uniquely amongst ABC transporter proteins, each subunit of an ABCI protein is encoded by a separate gene. Therefore, many ABCI protein sequences identified were removed from the alignments generated as they did not align with the NBD of other ABC-transporters. In the phylogenetic analysis of the retained sequences, 2 monophyletic ABCI clades were defined. One was sister to the monophyletic ABCA protein clade, while the other is sister to a monophyletic clade that includes ABCD and ABCB proteins. Each of the ABCI clades comprise sequences from *A. thaliana*, *O. sativa*, *M. polymorpha* and *C. paradoxa*. ABCI proteins-encoding genes in the genome of *C. paradoxa* and the 3 previously mentioned land plant species is evidence that they were present in the last common ancestor of the Archaeplastida. However, absence of ABCI protein-encoding genes in the genomes of *C.*

*merolae*, *C. reinhardtii* and *K. nitens* suggests that ABCI protein-encoding genes have been lost in these algal lineages. Furthermore, the tree topology shows that losses of ABCI proteins have occurred in the land plants. The absence of ABCI proteins-encoding genes in *A. agrestis*, *P. patens* and *S. moellendorffii* suggest a loss of ABCI proteins in the hornwort, moss and lycophyte lineages (Fig 2). However, since many putative ABCI proteins were removed from the analysis, we cannot be entirely certain that members of this subfamily have been entirely lost from these lineages mentioned. The evolution of the ABCI gene clade may be more difficult to resolve compared to other ABC transporter clades, due to their unique property of individual genes encoding each protein subunit.

Taken together the data show that ABC transporters are conserved among the Archaeplastida. There are sequences from the 10 sampled species from the Archaeplastida that constitute four previously identified monophyletic clades, B, C, E/F and G (Fig 1). This suggests that ABC transporters had diverged into these four subfamilies before the divergence of the Archaeplastida lineages and were therefore present in the last common ancestor of all species included in the analysis. We also identified 6 other ancient monophyletic clades of ABC transporter which constituted sequences that were present in either glaucophyte, or both glaucophyte and rhodophyte algae (Cp1, Cp2, Cp3, Cp4, Cp5 and CpCm1). ABCA, ABCD and ABCI proteins are not present in the genomes of every species of the Archaeplastida analysed here. Our phylogenetic analysis revealed that although ABCA proteins were present in the last common ancestor of the Archaeplastida, members of this clade were subsequently lost in the chlorophyte, hornwort, moss and lycophyte lineages. We show that ABCD proteins also had an ancient origin, likely already having diverged in the last common ancestor of the Archaeplastida. ABCD proteins were subsequently lost in the rhodophyte, chlorophyte, hornwort, moss and lycophyte lineages. The ABCI proteins included in our analysis comprise 2, distantly related clades. Conservation of ABCI proteins included in our analysis was limited, however, their presence in *C. paradoxa* suggests an ancient origin of ABCI proteins. No ABCI protein-encoding sequences were found in the rhodophyte, chlorophyte, streptophyte algae, hornwort, moss or lycophyte lineages, which is consistent with the loss of genes encoding these proteins from these lineages. This suggests that ABCI proteins are ancient but have been lost from many lineages during Archaeplastida evolution.

## Plant UGTs diversified recently, 14/20 identified groups are only present in Angiosperms

There are currently 45 known families of glycosyltransferases (GTs) in plants [60]. Uridine diphosphate glycosyltransferases (UGTs) are the largest glycosyltransferase family and are the only glycosyltransferase family to have been reported to contribute to NTSR. They comprise a monophyletic clade, known as Family 1 GTs. Family 28 GTs are a distinct glycosyltransferase protein family that are related to UGTs [23, 61–63]. Family 28 GTs have not been reported to contribute to NTSR. Family 28 GTs were included as an outgroup for our phylogenetic analysis. To investigate the evolutionary history of UGTs in Archaeplastida, phylogenetic trees were constructed using maximum likelihood statistics (Fig 3). This analysis showed that UGTs are only present in the streptophyte lineage of the Archaeplastida. UGTs are present in other eukaryotes, suggesting they were either lost in the glaucophyte, rhodophyte and chlorophyte lineages, or were introduced into the last common ancestor of the streptophyte lineage by horizontal gene transfer. Evidence of horizontal gene transfer of a glycosyltransferase gene has been previously reported from a Verrucomicrobium into the green alga *Picochlorum* SENEW3 [64]. Our phylogenetic analysis also shows UGT sequences comprise 20 major monophyletic groups, 16 of which were defined in previous analyses [25, 26].

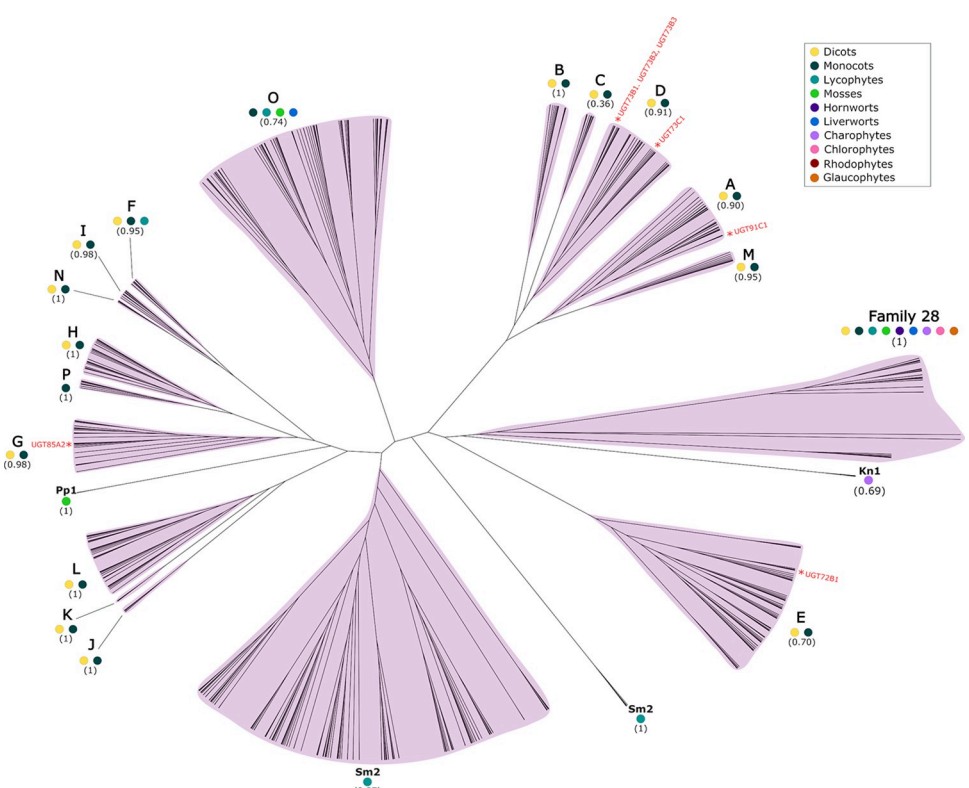

**Fig 3. Phylogenetic analysis of UGT protein sequences in the Archaeplastida.** Unrooted cladogram of a maximum likelihood (ML) analysis of Archaeplastida UGT proteins conducted by PhyML 3.0 [50] using an estimated gamma distribution parameter, the LG_G+F model of amino acid substitution, and a Chi$^2$-based approximate likelihood ratio (aLRT) test. Support values from the aLRT test for each clade are presented in parentheses. Protein sequences were aligned using MAFFT with the L-INS-I algorithm. UGT classes are indicated by pink highlighting and letters. Coloured dots indicate the presence of sequences from different species in each group. *A. thaliana* (yellow); *O. sativa* (dark green); *S. moellendorffii* (light blue); *P. patens* (light green); *A. agrestis* (dark purple); *M. polymorpha* (blue); *K. nitens* (light purple); *C. reinhardtii* (pink); *C. merolae* (maroon); *C. paradoxa* (orange). Confirmed NTSR genes or their *A. thaliana* homolog reported in Table 5 are indicated.

34 Family 28 GT sequences were identified in the genomes of the 10 Archaeplastida species, and these were included in the phylogenetic analysis. Family 28 GT sequences were found in 9 of the 10 Archaeplastida species but are missing from the rhodophyte alga *C. merolae*. This is consistent with Family 28 GTs being present in the last common ancestor of the Archaeplastida but subsequently lost in the rhodophyte algae lineage (Fig 4). This likely represents separate origins of Family 28 GTs and UGTs in the Archaeplastida, the latter of which is only present in the streptophyte lineage.

The UGT protein family has gone through many diversification events since the divergence of Family 28 GTs and UGTs from a last common ancestor. We identified 20 distinct monophyletic groups of UGTs, 16 of which had been previously identified. Previously reported groups are named alphabetically. Each of the 4 new groups identified in this analysis are comprised of UGT sequences from a single species and are named according to that species. The streptophyte algae *K. nitens* is the earliest diverging member of the Archaeplastida found with UGT protein-encoding genes in its genome. Three UGTs were identified in *K. nitens* and these formed a single monophyletic group (Kn1), sister to all other UGTs identified in the analysis. This is consistent with UGTs being present in the last common ancestor of the streptophytes since *K. nitens* is the earliest diverging member of the streptophyte lineage included

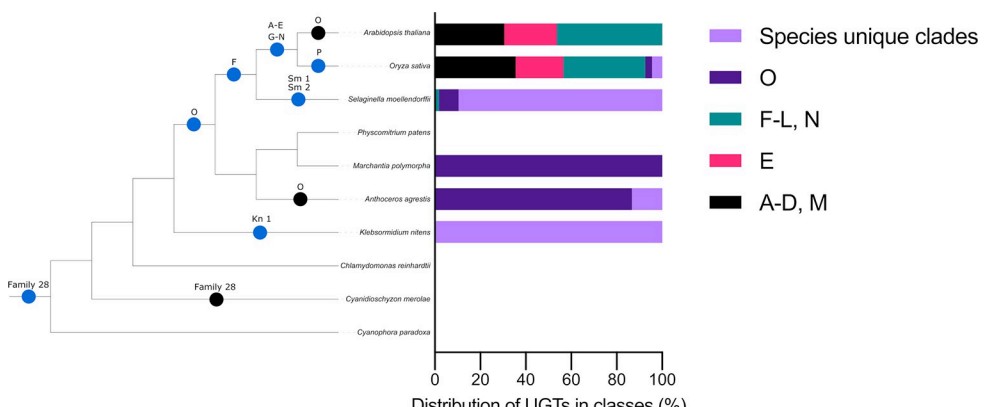

**Fig 4. UGTs are present in streptophytes and have diversified into many clades in the vascular lineage Angiosperms.** Cladogram of Archaeplastida phylogeny showing UGT clade origins and losses during plant evolution. The origins and losses of Family 28 glycosyltransferases are also shown. Blue circles represent first appearances of a clade, black circles represent the absence of a clade in a particular lineage. Also shown are the relative distributions of UGTs into their respective clades for each species included in our analysis, where each distinct coloured block represents either a separate UGT clade or a group of clades.

in this analysis. The group O clade comprises 83 UGTs from 4 species of the Archaeplastida– *O. sativa*, *S. moellendorffii*, *P. patens* and *M. polymorpha*. Conservation of these sequences in these 4 land plant species and their absence from the genomes of the streptophyte algae indicates that this clade originated after the divergence of land plants from the streptophyte algae. Group O UGTs sequences are not present in the genomes of *A. thaliana* and *A. agrestis*, suggesting a loss of genes in the eudicot lineage after the divergence of *O. sativa* and *A. thaliana* from their last common ancestor, and in the hornwort lineage respectively. UGTs sequences from *O. sativa*, *S. moellendorffii*, *P. patens*, *M. polymorpha* and also *A. thaliana* constitute a large monophyletic clade consisting of UGT protein from groups F, G, H, I, J, K, L, N, P and Pp1. This suggests that this clade originated at approximately the same stage as group O UGTs, and later diverged into each of the 10 groups [25, 26]. This evidence suggests few divergence events of UGT groups occurred during early streptophyte evolution and then more prolific diversification in more recently diverging lineages.

13 of the 20 UGT groups–A, B, C, D, E, G, H, I, J, K, L, M, N–are only present in the genomes of *A. thaliana* and *O. sativa*. This is consistent with these groups being present in the last common ancestor of the angiosperms (Fig 4). Since these sequences are not present in the genome of *S. moellendorffii*, it suggests that they originated after the divergence of the seed plant lineage from the lycophytes. There are group F UGT sequences in each of the 3 vascular plant species included in the analysis, suggesting this group originated in the last common ancestor of the vascular plant lineage. Group F is sister to groups N and I, which comprise only angiosperm sequences. This suggests groups N and I diverged from a common ancestor which they shared with group F. These data show most UGT diversification in extant taxa occurred after the divergence of the seed plants from the lycophytes.

Five groups were defined by sequences from a singles species in our analysis. Group P comprises UGT sequences from only the monocot angiosperm *O. sativa*. There are sequences from only the lycophyte *S. moellendorffii* in groups Sm1 and Sm2, sequences from only the streptophyte algae *K. nitens* in the group Kn1, and there are sequences only from the moss *P. patens* in group Pp1. The formation of these monospecific clades is consistent with recent origins of these sequences but could also be an artifact of poor taxon sampling in this analysis. Group P is only found in *O. sativa* and not the other angiosperm *A. thaliana*, suggesting that group P

originated after the divergence of the monocots and the eudicots or was lost in the dicot lineage. Group Sm1 is the largest clade identified in our analysis and comprises 141 UGT sequences. Sm1 sequences were only found in the *S. moellendorffii* genome. This is consistent with this group originating in the lycophyte lineage, following the divergence of lycophytes from other vascular plants. The large number of sequences in this group suggest a prolific expansion of the Sm1 UGTs in the lycophyte lineage. Sequences from group Sm2 were also identified only in *S. moellendorffii*, indicating this group also originated in the lycophyte lineage. However, with 6 sequences in this group, Sm2 did not go through the same extensive expansion as Sm1. Streptophyte algae sequences from *K. nitens* form a monophyletic clade (Kn1) which is sister to all other UGTs. Kn1 likely represents the earliest diverging UGTs identified in this analysis. Two moss sequences form a monophyletic clade (Pp1). Pp1 is sister to a larger monophyletic clade consisting of groups F, G, H, I, P and N. This suggests Pp1 genes share a common ancestor with sequences from groups F, G, H, I, P and N which that existed before the divergence of bryophytes from other land plants. However, the absence of Pp1 genes in *M. polymorpha* and *A. agrestis* suggest that sequences from this group were lost in these lineages. The origins and losses of these groups in different Archaeplastida lineages are summarised in Fig 4.

In summary, we show UGT sequences are present in streptophyte algae but not in glaucophytes, rhodophytes or chlorophytes; UGTs are encoded in the *K. nitens* genome and in land plants except the hornworts where we propose they were secondarily lost. This indicates that UGTs first arose after the divergence of the streptophyte lineage and continued to diversify following the colonisation of land.

## Six ABC-transporter sub-families have been associated with non-target site resistance to herbicides

ABC transporters have been implicated in non-target site herbicide resistance (NTSR) of crops and weeds. To identify the clades ABC transporters that are associated with NTSR belong, a literature survey was conducted. This survey identified genes that have been associated with NTSR because the mRNA levels of the ABC transporter mRNA are higher in resistant than sensitive populations. These genes are classified as "candidate NTSR genes" and are listed in (Table 2). ABC transporters demonstrated to increase herbicide resistance when expressed at high levels as a transgene or by other experimental methods are classified as "confirmed NTSR genes" and are listed in (Table 3).

We identified a total of 41 ABC transporter candidate NTSR genes in a variety of weed and non-weed species from the literature (Table 2). They include the grass weeds, Italian ryegrass (*Lolium multiflorum*) [65], red sprangletop (*Leptochloa chinensis*) [66], American slough grass (*Beckmannia syzigachne*) [67], shortawn foxtail (*Alopecurus aequalis*) [68] and jungle rice (*Echinochloa colona*) [69], the broad leaf (eudicot) weeds flixweed (*Descurainia Sophia*) [70], hairy fleabane (*Conyza bonariensis*) [71], giant chickweed (*Myosoton aquaticum*) [66, 72], and Palmer amaranth (*Amarnthus palmeri*) [73] and the crop rice (*Oryza sativa*) [74]. The candidate NTSR genes were identified in populations resistance to 5 different HRAC mode of action herbicide classes, including acetolactate synthase inhibitors, acetyl coA carboxylase inhibitors, enolpyruvyl shikimate phosphate synthase inhibitors, glutamine synthetase inhibitors and auxin mimic/cellulose synthesis inhibitors.

There are 12 monophyletic clades of ABC transporters (A, B, C, D, E/F, G, Cp1, Cp2, Cp3, Cp4, Cp5 and CpCm1). Candidate NTSR genes encoding ABC transporters are found in subfamilies ABCA, ABCB, ABCC, ABCF, ABCG and ABCI (Table 2). Thirteen ABCB and 18 ABCC members were identified as candidate NTSR genes. ABCG proteins constitute by far

**Table 2. Plant ABC transporter proteins that have been associated with resistance to herbicides are found within 6 classes identified in our analysis.**

| SUB-FAMILY | GENE NAME | EVIDENCE | HERBICIDE | HRAC CLASS | REFERENCE |
|---|---|---|---|---|---|
| **ABCA** | LmABCA7 (Italian ryegrass) | Up-regulated in NTSR plants compared to sensitive plants | glyphosate | 9 | [65] |
| **ABCB** | CbABCB1 (hairy fleabane) | Up-regulated in NTSR plants compared to sensitive plants | glyphosate | 9 | [71] |
| | MaABCB2 (giant chickweed) | Up-regulated in NTSR plants compared to sensitive plants | Tribenuron-methyl | 2 | [75] |
| | MaABCB2 (giant chickweed) | Up-regulated in NTSR plants compared to sensitive plants | Tribenuron-methyl | 2 | [72] |
| | LcABCB2 (red sprangletop) | Up-regulated in NTSR plants compared to sensitive plants | cyhalofop-butyl | 1 | [66] |
| | ApABCB2 (Palmer amaranth) | Up-regulated in NTSR plants compared to sensitive plants | glufosinate | 10 | [73] |
| | OsABCB4 (rice) | Up-regulated upon safener treatment which causes herbicide tolerance in rice | Fenoxaprop-P-ethyl | 1 | [74] |
| | BsABCB10 (American slough grass) | Up-regulated in NTSR plants compared to sensitive plants | fenoxaprop-p-ethyl | 1 | [67] |
| | AaABCB11 (shortawn foxtail) | Up-regulated in NTSR plants compared to sensitive plants | mesosulfuron-methyl | 2 | [68] |
| | OsABCB11 (rice) | Up-regulated upon safener treatment which causes herbicide tolerance in rice | Fenoxaprop-P-ethyl | 1 | [74] |
| | MaABCB19 (giant chickweed) | Up-regulated in NTSR plants compared to sensitive plants | Tribenuron-methyl | 2 | [72] |
| | CbABCB25 (hairy fleabane) | Up-regulated in NTSR plants compared to sensitive plants | glyphosate | 9 | [71] |
| | CbABCB27 (hairy fleabane) | Up-regulated in NTSR plants compared to sensitive plants | glyphosate | 9 | [71] |
| | MaABCB29 (giant chickweed) | Up-regulated in NTSR plants compared to sensitive plants | Tribenuron-methyl | 2 | [75] |
| **ABCC** | DsABCC1 (flixweed) | Up-regulated in NTSR plants compared to sensitive plants | tribenuron-methyl | 2 | [70] |
| | CbABCC2 (hairy fleabane) | Up-regulated in NTSR plants compared to sensitive plants | glyphosate | 9 | [71] |
| | MaABCC3 (giant chickweed) | Up-regulated in NTSR plants compared to sensitive plants | Tribenuron-methyl | 2 | [75] |
| | MaABCC3 (giant chickweed) | Up-regulated in NTSR plants compared to sensitive plants | Tribenuron-methyl | 2 | [72] |
| | CbABCC4 (hairy fleabane) | Up-regulated in NTSR plants compared to sensitive plants | glyphosate | 9 | [71] |
| | AaABCC8 (shortawn foxtail) | Up-regulated in NTSR plants compared to sensitive plants | mesosulfuron-methyl | 2 | [68] |
| | MaABCC8 (giant chickweed) | Up-regulated in NTSR plants compared to sensitive plants | Tribenuron-methyl | 2 | [75] |
| | CbABCC8 (hairy fleabane) | Up-regulated in NTSR plants compared to sensitive plants | glyphosate | 9 | [71] |
| | LcABCB8 (red sprangletop) | Up-regulated in NTSR plants compared to sensitive plants | cyhalofop-butyl | 1 | [66] |
| | CbABCC9 (hairy fleabane) | Up-regulated in NTSR plants compared to sensitive plants | glyphosate | 9 | [71] |
| | CbABCC10 (hairy fleabane) | Up-regulated in NTSR plants compared to sensitive plants | glyphosate | 9 | [71] |
| | MaABCC10 (giant chickweed) | Up-regulated in NTSR plants compared to sensitive plants | Tribenuron-methyl | 2 | [75] |
| | MaABCC10 (giant chickweed) | Up-regulated in NTSR plants compared to sensitive plants | Tribenuron-methyl | 2 | [72] |
| | OsABCB10 (rice) | Up-regulated upon safener treatment which causes herbicide tolerance in rice | Fenoxaprop-P-ethyl | 1 | [74] |
| | CbABCC12 (hairy fleabane) | Up-regulated in NTSR plants compared to sensitive plants | glyphosate | 9 | [71] |
| | CbABCC13 (hairy fleabane) | Up-regulated in NTSR plants compared to sensitive plants | glyphosate | 9 | [71] |
| | CbABCC14 (hairy fleabane) | Up-regulated in NTSR plants compared to sensitive plants | glyphosate | 9 | [71] |
| | LcABCB15 (red sprangletop) | Up-regulated in NTSR plants compared to sensitive plants | cyhalofop-butyl | 1 | [66] |
| **ABCF** | DsABCF5 (flixweed) | Up-regulated in NTSR plants compared to sensitive plants | tribenuron-methyl | 2 | [70] |
| **ABCG** | CbABCG7 (hairy fleabane) | Up-regulated in NTSR plants compared to sensitive plants | glyphosate | 9 | [71] |
| | MaABCG8(giant chickweed) | Up-regulated in NTSR plants compared to sensitive plants | Tribenuron-methyl | 2 | [72] |
| | OsABCG11 (rice) | Up-regulated upon safener treatment which causes herbicide tolerance in rice | Fenoxaprop-P-ethyl | 1 | [74] |
| | MaABCG14 (giant chickweed) | Up-regulated in NTSR plants compared to sensitive plants | Tribenuron-methyl | 2 | [75] |
| | MaABCG14 (giant chickweed) | Up-regulated in NTSR plants compared to sensitive plants | Tribenuron-methyl | 2 | [72] |
| | OsABCG39 (rice) | Up-regulated upon safener treatment which causes herbicide tolerance in rice | Fenoxaprop-P-ethyl | 1 | [74] |
| | EcABCG48 (jungle rice) | regulated in NTSR plants compared to sensitive plants | quinclorac | 4/29 | [69] |
| **ABCI** | CbABCI17 (hairy fleabane) | Up-regulated in NTSR plants compared to sensitive plants | glyphosate | 9 | [71] |

**Table 3. Plant ABC transporter proteins that have been empirically shown to confer resistance or tolerance to xenobiotics are found in 3 classes identified in our analysis.**

| SUB-FAMILY | GENE NAME | EVIDENCE | HERBICIDE/XENOBIOTIC | HRAC CLASS | REFERENCE |
|---|---|---|---|---|---|
| ABCC | AtABCC1 (*A. thaliana*) | Show this gene encodes a protein capable of transporting GS-conjugated metolachlor | metolachlor | 15 | [76] |
| ABCC | EcABCC8 (jungle rice) | Causes resistance when overexpressed in transgenic rice | glyphosate | 9 | [77] |
| ABCB | AtABCB1 (*A. thaliana*) | Endogenous overexpression gives resistance to multiple herbiciides | dicamba, oryzalin, pendimethalin, MSMA | 4, 3, 3, 0 | [78] |
| ABCG | AtABCG19 (*A. thaliana*) | Endogenous overexpression gives resistance | kanamycin | N/A | [79] |
| ABCG | AtABCG37 (*A. thaliana*) | GOF mutant more resistant, LOF mutant more sensitive | 2,4-D | 4 | [7] |
| ABCG | AtABCG39 (*A. thaliana*) | LOF mutant is resistant | paraquat | 22 | [8] |

The position of these genes or their *A. thaliana* homologs has been indicated in the phylogeny in Fig 1.

the largest monophyletic clade we identified in our phylogenetic analysis, representing approximately 2/3 of all the ABC transporters included. However, relatively few ABCG transporters were identified as candidate NTSR genes. From 4 species, we identified 6 distinct ABCG candidate NTSR genes–ABCG7, ABCG8, ABCG11, ABCG14, ABCG39 and ABCG48. A single candidate NTSR gene was found in the literature from each of the groups ABCA, ABCF and ABCI. Taken together, these data suggest candidate NTSR genes encoding ABC transporters belong to clades that were present in the last common ancestor of the Archaeplastida and that the frequency of candidate NTSR genes is not proportional to ABC transporter clade size.

Six ABC transporter proteins have been experimentally confirmed to confer NTSR in plants (Table 3). These 6 genes belong to 3 different clades–ABCB, ABCC and ABCG. These clades comprise sequences from all 10 species of Archaeplastida included in the analysis, and therefore had already diverged in the last common ancestor of the Archaeplastida. It can be concluded that ABC transporters that have been confirmed to contribute to NTSR belong to clades which were already present in the common ancestor of the Archaeplastida. The ABC transporters confirmed to confer resistance to herbicides did so to a variety of HRAC mode of action groups, including very long-chain fatty acid inhibitors, enolpyruvyl shikimate phosphate synthase inhibitors, auxin mimics, microtubule assembly inhibitors, PSI inhibitors, and a herbicide with an unknown mode of action (MSMA). We therefore show that ABC transporter proteins within the same monophyletic clade confer resistance to herbicides with different modes of action.

## UGTs groups associated with herbicide resistance evolved after the divergence of Angiosperms from the Archaeplastida lineage

Many UGT genes have also been reported to contribute to NTSR in populations of resistant weeds in the field. We identified a total of 34 candidate NTSR UGT genes in a variety of weed and non-weed species from the literature (Table 4). They include the grass weeds, Italian ryegrass (*Lolium multiflorum*) [65], American slough grass (*Beckmannia syzigachne*) [67, 74, 80], *Brachypodium hybridum* [81], shortawn foxtail (*Alopecurus aequalis*) [68], Tausch's goat grass (*Aegilops tauschii*) [82], and jungle rice (*Echinochloa colona*) [69], the broad leaf (eudicot)

**Table 4. Plant UGTs associated with herbicide resistance are found in 11 classes we have identified in our analysis.**

| GROUP | FAMILY + SUB-FAMILY | GENE NAME | EVIDENCE | HERBICIDE | HRAC CLASS | REFERENCE |
|---|---|---|---|---|---|---|
| A | 79 | LmUGT79 (Italian ryegrass) | Up-regulated in NTSR plants compared to sensitive plants | glyphosate | 9 | [65] |
| | 79 | OsUGT79 (rice) | Up-regulated upon safener treatment which causes herbicide tolerance in rice | Fenoxaprop-P-ethyl | 1 | [74] |
| | 79B | DsUGT79B6 (flixweed) | Up-regulated in NTSR plants compared to sensitive plants | tribenuron-methyl | 2 | [70] |
| | 91A | DsUGT91A1 (flixweed) | Up-regulated in NTSR plants compared to sensitive plants | tribenuron-methyl | 2 | [70] |
| B | 89A | CbUGT89A (hairy fleabane) | Up-regulated in NTSR plants compared to sensitive plants | glyphosate | 9 | [71] |
| C | 90A | BsUGT90A1 (American slough grass) | Up-regulated in NTSR plants compared to sensitive plants | fenoxaprop-p-ethyl | 1 | [67] |
| | 71C | DsUGT71C2 (flixweed) | Up-regulated in NTSR plants compared to sensitive plants | tribenuron-methyl | 2 | [70] |
| D | 73B | MaUGT73B3 (giant chickweed) | Up-regulated in NTSR plants compared to sensitive plants | tribenuron-methyl | 2 | [75] |
| | 73B | BhUGT73B5 (*Brachypodium hybridum*) | Up-regulated upon herbicide application | pinoxaden | 1 | [81] |
| | 73B | AaUGT73B5 (shortawn foxtail) | Up-regulated in NTSR plants compared to sensitive plants | mesosulfuron-methyl | 2 | [68] |
| | 73C | AtUGT73C (Tausch's goat grass) | Up-regulated in NTSR plants compared to sensitive plants | mesosulfuron-methyl | 2 | [82] |
| | 73C | BsUGT73C1 (American slough grass) | Up-regulated in NTSR plants compared to sensitive plants | fenoxaprop-p-ethyl | 1 | [80] |
| | 73C | AaUGT73C5 (shortawn foxtail) | Up-regulated in NTSR plants compared to sensitive plants | mesosulfuron-methyl | 2 | [68] |
| | 73D | EcUGT73D1 (jungle rice) | Up-regulated in NTSR plants compared to sensitive plants | quinclorac | 4/29 | [69] |
| | 73E | CbUGT73E1 (hairy fleabane) | Up-regulated in NTSR plants compared to sensitive plants | glyphosate | 9 | [71] |
| E | 71B | DsUGT71B1 (flixweed) | Up-regulated in NTSR plants compared to sensitive plants | tribenuron-methyl | 2 | [70] |
| | 88A | BhUGT88A1 (*Brachypodium hybridum*) | Up-regulated upon herbicide application | pinoxaden | 1 | [81] |
| F | 78D | MaUGT78D2 (giant chickweed) | Up-regulated in NTSR plants compared to sensitive plants | tribenuron-methyl | 2 | [75] |
| | 78D | MaUGT78D2 (giant chickweed) | Up-regulated in NTSR plants compared to sensitive plants | tribenuron-methyl | 2 | [72] |
| | 78D | DsUGT78D4 (flixweed) | Up-regulated in NTSR plants compared to sensitive plants | tribenuron-methyl | 2 | [70] |
| G | 85A | BhUGT85A2 (*Brachypodium hybridum*) | Up-regulated upon herbicide application | pinoxaden | 1 | [81] |
| | 85A | BsUGT85A2 (American slough grass) | Up-regulated in NTSR plants compared to sensitive plants | fenoxaprop-p-ethyl | 1 | [80] |
| | 85C | CbUGT85C1 (hairy fleabane) | Up-regulated in NTSR plants compared to sensitive plants | glyphosate | 9 | [71] |
| | 85C | CbUGT85C2 (hairy fleabane) | Up-regulated in NTSR plants compared to sensitive plants | glyphosate | 9 | [71] |
| H | 76B | CbUGT76B1 (hairy fleabane) | Up-regulated in NTSR plants compared to sensitive plants | glyphosate | 9 | [71] |
| I | 83A | AaUGT83A1 (shortawn foxtail) | Up-regulated in NTSR plants compared to sensitive plants | mesosulfuron-methyl | 2 | [68] |

*(Continued)*

**Table 4.** (Continued)

| GROUP | FAMILY + SUB-FAMILY | GENE NAME | EVIDENCE | HERBICIDE | HRAC CLASS | REFERENCE |
|---|---|---|---|---|---|---|
| | 83A | BsUGT83A1 (American slough grass) | Up-regulated in NTSR plants compared to sensitive plants | fenoxaprop-p-ethyl | 1 | [67] |
| | 83A | OsUGT83A1 (rice) | Up-regulated upon safener treatment which causes herbicide tolerance in rice | fenoxaprop-P-ethyl | 1 | [74] |
| L | 74D | DsUGT74D1 (flixweed) | Up-regulated in NTSR plants compared to sensitive plants | tribenuron-methyl | 2 | [70] |
| | 75D | AaUGT75D1 (shortawn foxtail) | Up-regulated in NTSR plants compared to sensitive plants | mesosulfuron-methyl | 2 | [68] |
| | 75D | EcUGT75D1 (jungle rice) | Up-regulated in NTSR plants compared to sensitive plants | quinclorac | 4/29 | [69] |
| | 84A | DsUGT84A1 (flixweed) | Up-regulated in NTSR plants compared to sensitive plants | tribenuron-methyl | 2 | [70] |
| M | 92A | AaUGT92A1 (shortawn foxtail) | Up-regulated in NTSR plants compared to sensitive plants | mesosulfuron-methyl | 2 | [68] |
| UGT13248 | | AtUGT13248 (Tausch's goat grass) | Up-regulated in NTSR plants compared to sensitive plants | mesosulfuron-methyl | 2 | [82] |

weeds flixweed (*Descurainia Sophia*) [70], hairy fleabane (*Conyza bonariensis*) [71], and giant chickweed (*Myosoton aquaticum*) [66, 72] and the crop rice (*Oryza sativa*) [74]. The candidate NTSR genes were identified in populations resistant to 4 classes of HRAC mode of action herbicides, including acetolactate synthase inhibitors, acetyl coA carboxylase inhibitors, enolpyruvyl shikimate phosphate synthase inhibitors and auxin mimic/cellulose synthesis inhibitors.

There are 16 monophyletic groups of UGT sequences in angiosperms. We identified candidate NTSR genes from 11 of these monophyletic groups in angiosperm weeds and crops (Table 4). There are 8 candidate NTSR group D UGT genes in 7 different species. These represent 7 distinct genes, UGT73B3, UGT73B5, UGT73C, UGT73C1, UGT73C5, UGT73D1 and UGT73E1. UGT73B5 was identified in both *B. hybridum* and *A. aequalis*. A single candidate

**Table 5. Plant UGTs that have been empirically shown to confer resistance in plants belong to 4 classes identified in our analysis.**

| GROUP | FAMILY + SUB-FAMILY | GENE NAME | EVIDENCE | HERBICIDE/XENOBIOTIC | HRAC CLASS | REFERENCE |
|---|---|---|---|---|---|---|
| A | 91C | AtUGT91C1 (*A. thaliana*) | Demonstrated glucosylation of sulcotrione and herbicide resistance in overexpression lines | sulcotrione | 27 | [83] |
| E | 72B | AtUGT72B1 (*A. thaliana*) | Knockout plants no longer conjugated these chemicals | 3,4-dichloroaniline (DCA) and 2,4,5-trichlorophenol (TCP) | N/A | [84] |
| D | 73C | BsUGT73C1 (American slough grass) | Identified in RNA-seq experiment–causes resistance when expressed in *B. distachyon* | fenoxaprop-p-ethyl | 1 | [80] |
| G | 85A | BsUGT85A2 (American slough grass) | Identified in RNA-seq experiment–causes resistance when expressed in *B. distachyon* | fenoxaprop-p-ethyl | 1 | [80] |
| D | 73B | AtUGT73B1 (*A. thaliana*) | Mutations leading to overexpression cause herbicide resistance | paraquat | 22 | [85] |
| D | 73B | AtUGT73B2 (*A. thaliana*) | Mutations leading to overexpression cause herbicide resistance | paraquat | 22 | [85] |
| D | 73B | AtUGT73B3 (*A. thaliana*) | Mutations leading to overexpression cause herbicide resistance | paraquat | 22 | [85] |

The position of these genes or their *A. thaliana* homologs has been indicated in the phylogeny in Fig 3.

NTSR gene was identified in flixweed, UGT71B1, a group E UGT. Group E is the second largest UGT group. No candidate NTSR UGT sequences were identified from group O, which is the largest UGT group in which angiosperm sequences were identified. These data indicate UGT proteins that detoxify herbicides belong to clades that were present in the last common ancestor of the Angiosperms.

Seven UGT genes, from 4 groups–A, D, E and G–have been experimentally confirmed to confer NTSR in plants (Table 5). There are genes from only two species, the angiosperms *A. thaliana* and *O. sativa*, in each of these groups in our phylogenetic analysis. This suggests that these groups were present in the last common ancestor and evolved following the divergence of the angiosperms from the lycophytes. 6 of the 7 genes identified confer NTSR to 3 HRAC mode of action classes, HPPD inhibitors, acetyl coA inhibitors and PSI inhibitors.

## Discussion

Herbicide use has increased continuously since the 1940s, leading to the evolution of herbicide resistance in weeds through selection of naturally occurring alleles that confer herbicide resistance. Overexpression of genes encoding some members of the ATP-binding cassette (ABC) transporter family and uridine diphosphate (UDP) glycosyltransferase (UGT) family in weeds confers herbicide non-target site resistance (NTSR) [2, 4, 5]. ABC transporters are ATP-driven pumps that transport molecules across membranes, while UGT enzymes catalyse the metabolism of a diversity of compounds by transferring a sugar moiety to an acceptor molecule. By carrying out phylogenetic analyses, we set out to resolve the evolutionary relationships of these protein families, and to classify those genes associated with NTSR. We compared sequences from 10 species which represent distinct lineages of the Archaeplastida and generated phylogenetic trees. From the topology of these trees, it can be concluded that ABC transporters were present in the last common ancestor of the Archaeplastida. There are 12 monophyletic clades of ABC transporters (A, B, C, D, E/F, G, Cp1, Cp2, Cp3, Cp4, Cp5, and CpCm1) and one polyphyletic clade (I), all of which existed in the last common ancestor of the Archaeplastida. UGT enzymes existed in the last common ancestor of the streptophytes and have since diversified into 20 monophyletic clades (A, B, C, D, E, F, G, H, I, J, K, L, M, N, O, P, Kn1, Pp1, Sm1, and Sm2). Overall, the phylogenetic analysis shows that ABC transporters were present in the last common ancestor of the Archaeplastida, while UGTs proteins are only present in the streptophyte lineage among the Archaeplastida.

In plants, ABC transporters transport of a variety of molecules across membranes [10, 80]. ABC transporters confer herbicide resistance by sequestering herbicides and their metabolites in the vacuole or extracellular spaces, where they cannot interact with their target [16]. We hypothesised that a high diversity of ABC transporters among the Archaeplastida may facilitate the high diversity of substrates which can be recognised and transported by ABC transporters. However, we have shown that closely related ABC transporters, within the same clade, can be associated with resistance to a diverse range of herbicide mode of actions. For example, proteins from clade A are associated with resistance to 6 different herbicides, from 4 different HRAC mode of action groups (HRAC group 1, 2, 9 and 10). This suggests that ABC transporter diversity is not necessary for a diversity of molecules which can be recognised and transported. Nonetheless, this diversity may play a role diversity of ABC transporter localisation, and hence in herbicide detoxification by allowing ABC transporters to sequester chemicals in various locations. These locations include the vacuole and extracellular spaces, where herbicides can be sequestered and are no longer toxic to the plant.

ABC transporter clades in extant plants are ancient and we show that the common ancestor of the Archaeplastida, which existed around 1.6 billion years ago in the Proterozoic era,

contained all clades of ABC transporters identified in our analysis [38, 86, 87]. These ABC transporters evolved during the Proterozoic to carry out various cellular functions, unrelated to herbicide resistance. However, certain ABC transporter genes are now being selected in modern agricultural systems for a new purpose–conferring resistance to herbicides. For example, ABCB1, a P-glycoprotein, which has been shown to cause resistance to dicamba, oryzalin, pendimethalin and MSMA when endogenously overexpressed in *A. thaliana* has been characterised [78]. ABCB1 has been shown to be involved in the transport of auxin, a crucial plant hormone essential for numerous developmental processes. ABCB1 mediates auxin efflux, regulating its distribution and concentration in different plant tissues. ABCB1 mutants have altered auxin distribution and as a result, show various developmental defects [88–90]. Herbicide resistance evolving through the altered function of genes that evolved to carry out an entirely different function, can be considered exaptation [91]. Therefore, we argue that the involvement of ABC transporters in herbicide resistance is an example of exaptation.

UGT enzymes catalyse the conjugation of sugars moieties to a variety of acceptor molecules, including herbicides which can lead to detoxification [22]. High sequence diversity among UGTs genes leads to protein structural heterogeneity, which in turn facilitates a wide range of acceptor molecules, including herbicides [24]. We have shown that a large proportion of UGT gene diversity evolved in the angiosperm lineage, and this may account for the ability of UGTs to detoxify herbicides. However, we have also shown that UGTs from closely related clades, or even those from within the same clade, may confer resistance to herbicides with a variety of different chemistries and modes of action. For example, UGT proteins from group D have been associated with, or confer resistance to, 7 different herbicides from 5 different HRAC mode of action groups (HRAC group 1, 2, 4/29, 9 and 22). This suggests sufficient variability has also evolved within UGT clades in extant plants to give UGTs the capacity to conjugate sugars to a diverse range of acceptor molecules, which includes herbicides.

UGTs proteins were present in the last common ancestor of the streptophytes, which existed 1.1–1.2 billion years ago, in the Proterozoic era [92–94]. However, UGTs are not present in the earlier diverging glaucophyte, rhodophyte or chlorophyte lineages. Since UGTs are present in other eukaryotes and some bacteria, their absence in non-streptophyte algal lineages suggests either a loss of UGT genes in these lineages or that UGTs were acquired in the last common ancestor of the streptophyte lineage through horizontal gene transfer. Our data suggest that there was little diversification of UGT sequences until after the vascular plants diverged from other streptophytes in the Silurian period of the Paleozoic era (420–450 million years ago) [92–95]. 17 of the 20 UGT clades we identified, evolved in the vascular lineage. 14 of these groups are present only in angiosperms, suggesting they evolved after the divergence of seed plants from lycophytes, which occurred in the Devonian period of the Paleozoic era (360–400 million years ago) [22, 92]. This diversification of UGTs likely accompanied an increase in morphological complexity of plants and the colonisation of a wider range of terrestrial habitats. UGTs were likely selected to carry out a variety of cellular functions in streptophytes, during their colonisation of land, and subsequent evolution of the vascular plants. Over the last 50 years, they have been selected for herbicide resistance in the agricultural environment.

We have shown that both ABC transporters and UGT enzymes evolved in the Proterozoic era. ABC transporters genes were present in the last common ancestor of the Archaeplastida and had already diversified into many clades in this lineage. Conversely, our data shows that UGTs genes are only present in the streptophyte lineage and diversified mainly in the vascular lineage among plants. The clade diversity of ABC transporters and UGT enzymes in extant plants may contribute to their ability to confer herbicide resistance. We propose that ABC transporters and UGTs were selected to carry out a variety of cellular functions in the Proterozoic, and now some members of these protein families are being selected to carry out an

entirely different function–conferring resistance to herbicides. ABC transporters and UGTs conferring herbicide resistance is therefore an example of exaptation.

## Supporting information

**S1 Fig. Protein features of ABC transporters and UDP-glycosyltransferases in plants.** Diagrams of the typical ABC transporter (A) and UDP-glycosyltransferases (B). A shows the domain architecture of ABC transporters from each of the major sub-families in plants. The diagram is adapted from https://doi.org/10.1038/srep16724. B shows α-helices (green), β-strands (purple) and residues reported to form part of the acceptor pocket (red dashed lines). The PSPG motif reported to interact with the UDP-sugar donor is highlighted in grey. The structure is based on that of AtUGT72B1 reported in https://doi.org/10.1016/j.phytochem.2008.12.009.
(TIF)

**S2 Fig. Number of ABC proteins identified in each clade.**
(TIF)

**S3 Fig. Number of UGT proteins identified in each clade.** Number of Family 28 glycosyltransferases identified in each species is also given.
(TIF)

**S1 Text. Untrimmed alignment of all ABC transporter sequences used in the phylogenetic analysis.**
(TXT)

**S2 Text. Untrimmed alignment of all UGT sequences used in the phylogenetic analysis.**
(TXT)

**S3 Text. Manually trimmed alignment of all ABC transporter sequences used in the phylogenetic analysis.**
(TXT)

**S4 Text. Manually trimmed alignment of all UGT sequences used in the phylogenetic analysis.**
(TXT)

## Author Contributions

**Writing – original draft:** Samuel Caygill.

**Writing – review & editing:** Liam Dolan.

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
