## [Decision Letter · Decision Letter 0]

27 Jun 2023

PONE-D-23-17239ATP binding cassette transporters and uridine diphosphate glycosyltransferases are ancient protein families that evolved roles in herbicide resistance through exaptationPLOS ONE

Dear Dr. Dolan,

Thank you for submitting your manuscript to PLOS ONE. After careful consideration, we feel that it has merit but does not fully meet PLOS ONE’s publication criteria as it currently stands. Therefore, we invite you to submit a revised version of the manuscript that addresses the points raised during the review process.

We look forward to receiving your revised manuscript.

Kind regards,

Yanbin Yin

Academic Editor

PLOS ONE

Journal Requirements:

Reviewers' comments:

Reviewer's Responses to Questions

**Comments to the Author**

1. Is the manuscript technically sound, and do the data support the conclusions?

Reviewer #1: Yes

Reviewer #2: Yes

2. Has the statistical analysis been performed appropriately and rigorously? 

Reviewer #1: Yes

Reviewer #2: Yes

3. Have the authors made all data underlying the findings in their manuscript fully available?

Reviewer #1: Yes

Reviewer #2: Yes

4. Is the manuscript presented in an intelligible fashion and written in standard English?

Reviewer #1: Yes

Reviewer #2: Yes

5. Review Comments to the Author

Reviewer #1: The authors presented a study on evolution and diversification of ABC transporters and UGTs Archaeplastida. This is a novel and interesting report. The data are novel and largely support the conclusions. I have some comments as follows:

1. In the ‘Data Resources’, it seems that there is no information about S. moellendorffii, please add the information.

2. Line 355-359, the authors said that UGTs were only present in streptophyte and suggested they were gained by horizontal gene transfer, I wonder if the authors have any evidence for the HGT, or from bacteria, fungi or other organisms?

Reviewer #2: Table 1 Here you have listed moss as Physcomitrella patens, however in the Figures you have Physcomitrium patens, please change accordingly

Line 126 UGTs are present in the streptophyte lineage; it reads better with the word in before the streptophyte lineage

Data Resources

Only 9 out of 10 species are listed. You forgot to mention where you retrieved the sequence from S. moellendorffii.

Line 181; What are the support values or bootstrap values of the phylogenetic trees.

Line 195; in order to stay consistent throughout the manuscript, I am recommending to use streptophyte alga instead of charophyte alga.

Fig 2 legend. Please state which colored blocks refers to which ABC family. I can only assume that ABCA is the black colored block.

How come there are two black colored blocks for C. paradoxa?

I would suggest to add a representative of the streptophyte algae ZCC-clade. It is postulated that Zygnematophyceae are the closest cousin to land plants, it would be interesting to see if a divergence of ABC-transporters and UTGs also took place in the ZCC-clade.

Fig 4 legend. Also, here please add which colored block belongs to which UGT family.

Fig 4. It is not clear that Family 28 is lost/not present in C.paradoxa and C. reinhardtii.

Line 419. Five instead of Five

Line 448. I do not agree with your statement that UGTs diversified after land colonization as you do not have other streptophytes in your analysis. I would rephrase the conclusion.

Line 486. A reference to a Figure or Table would be good to underline your statement. It would be good to add this information to the phylogenetic tree.

Line 551. A depiction of your UGTs that are confirmed NTSRs in the phylogeny would

6. PLOS authors have the option to publish the peer review history of their article (what does this mean?). If published, this will include your full peer review and any attached files.

Reviewer #1: No

Reviewer #2: **Yes: **Elisabeth Fitzek-Campbell

---

## [Author Response · Author response to Decision Letter 0]

26 Jul 2023

Dear Editor and Reviewers,

We would like to express our thanks to the reviewers for their valuable comments and suggested revisions for our submitted manuscript entitled “ATP binding cassette transporters and uridine diphosphate glycosyltransferases are ancient protein families that evolved roles in herbicide resistance through exaptation”. We have carefully considered each of the comments and made what we believe to be the necessary revisions to address them. Below we present our point-by-point response to the reviewer’s comments:

Reviewer #1:

1. In the ‘Data Resources’, it seems that there is no information about S. moellendorffii, please add the information.

Response: We apologise for the oversight in the initial submission. We have now included the information about S. moellendorffii in the ‘Data Resources’ section of the manuscript. We obtained the sequence data for S. moellendorffii from the Phytozome 12 database (https://phytozome.jgi.doe.gov/pz/portal.html).

2. Line 355-359, the authors said that UGTs were only present in streptophyte and suggested they were gained by horizontal gene transfer, I wonder if the authors have any evidence for the HGT, or from bacteria, fungi or other organisms?

Response: We appreciate the reviewer’s question regarding the evidence for horizontal gene transfer (HGT) of UGTs. While we do not have direct evidence of HGT from bacteria, fungi, or other organisms, we have added an example of HGT of a glycosyltransferase into an algal species in order to back up the rationale behind this hypothesis. The added example reads: ‘Evidence of horizontal gene transfer of a glycosyltransferase gene has been previously reported from a Verrucomicrobium into the green alga Picochlorum SENEW3.’ The reference for this statement is as follows:

 Foflonker F, Price DC, Qiu H, Palenik B, Wang S, Bhattacharya D. Genome of the halotolerant green alga P icochlorum sp. reveals strategies for thriving under fluctuating environmental conditions. Environ Microbiol [Internet]. 2015 Feb 1 [cited 2023 Jul 2];17(2):412–26. Available from: https://onlinelibrary.wiley.com/doi/10.1111/1462-2920.12541

Reviewer #2:

1. Table 1 Here you have listed moss as Physcomitrella patens, however in the Figures you have Physcomitrium patens, please change accordingly.

Response: We appreciate the reviewer’s keen observation. We have corrected the inconsistency in the nomenclature. We now refer to the moss species consistently as Physcomitrium patens in both the table and the figures throughout the manuscript.

2. Line 126 UGTs are present in the streptophyte lineage; it reads better with the word in before the streptophyte lineage.

Response: Thank you for the suggestion. We have made the recommended change to improve the clarity of the sentence. It now reads: ‘On the other hand, we show that UGTs are present in the streptophyte lineage of the Archaeplastida but absent from non-streptophyte lineages.’

3. Data Resources

Only 9 out of 10 species are listed. You forgot to mention where you retrieved the sequence from S. moellendorffii.

Response: We apologise for the oversight in listing the data resources. As mentioned in our response to Reviewer #1, we obtained the sequence data for S. moellendorffii from the Phytozome 12 database (https://phytozome.jgi.doe.gov/pz/portal.html). We have now included this information in the ‘Data Resources’ section of the manuscript.

4. Line 181; What are the support values or bootstrap values of the phylogenetic trees.

Response: We appreciate the reviewer’s request for the support values or bootstrap values of the phylogenetic trees. The support values were not initially included in the manuscript, but upon revisiting the data, we have now added the support values to Figure 1 and Figure 3. The support values from a Chi2-based approximate likelihood ratio (aLRT) test have been included for each of the major clades identified. We apologise for their initial omission.

5. Line 195; in order to stay consistent throughout the manuscript, I am recommending to use streptophyte alga instead of charophyte alga.

Response: We thank the reviewer for the suggestion. To ensure consistency throughout the manuscript, we have made the recommended change, and ‘streptophyte alga(e)’ is now used in place of ‘charophyte alga(e)’ throughout.

6. Fig 2 legend. Please state which colored blocks refers to which ABC family. I can only assume that ABCA is the black colored block.

How come there are two black colored blocks for C. paradoxa?

Response: We apologise for the lack of clarity and legend in Figure 2 and 4. To address this concern, we have added a legend to both of these figures. For Figure 4, some UGT clades had to be pooled to maintain clarity. However, we believe that this still gets our take home message across that angiosperm UGTs have diverged into many clades whereas the UGTs from other members of the streptophyte lineage have not. This has also allowed us to avoid duplicate colours.

7. I would suggest to add a representative of the streptophyte algae ZCC-clade. It is postulated that Zygnematophyceae are the closest cousin to land plants, it would be interesting to see if a divergence of ABC-transporters and UTGs also took place in the ZCC-clade.

Response: We appreciate the reviewer’s suggestion to include a representative of the streptophyte algae ZCC-clade. While we recognise the importance of studying the Zygnematophyceae in the context of the phylogeny of ABC transporters and UGTs, we must note that the main focus of our study was on the evolution and diversification of these gene families in the Archaeplastida, with a particular emphasis on the land plants. Including the Zygnematophyceae would require additional analyses and data, which extend beyond the scope of the current study. However, we acknowledge the significance of investigating the ZCC-clade and will consider its conclusion in future research.

8. Fig 4 legend. Also, here please add which colored block belongs to which UGT family.

Fig 4. It is not clear that Family 28 is lost/not present in C.paradoxa and C. reinhardtii.

Response: We apologise for the lack of clarity in regards to the figure legend and this has been addressed as described in our response to point 6. We apologise for the ambiguity regarding Family 28 in C. paradoxa and C. reinhardtii. Family 28 glycosyltransferase genes are present in the genomes of both of these species. We have included a note in the figure legend which we hope improves the clarity of this.

9. Line 419. Five instead of Five.

Response: Thank you for pointing out the typographical error. We have corrected the error to read ‘Five’ instead of ‘FIve’.

10. Line 448. I do not agree with your statement that UGTs diversified after land colonization as you do not have other streptophytes in your analysis. I would rephrase the conclusion.

Response: We appreciate the reviewer’s perspective on the statement regarding UGT diversification after land plant colonisation. Upon reconsideration, we agree that the conclusion could be better phrased to account for the absence of other streptophytes in our analysis. We have revised the conclusion to state that the findings show a diversification of UGTs in the sampled streptophytes which diverged after the colonisation of land. We no longer argue that this is linked to the colonisation of land.

11. Line 486. A reference to a Figure or Table would be good to underline your statement. It would be good to add this information to the phylogenetic tree.

Response: We thank the reviewer for suggesting the inclusion of a reference to a Figure or Table to support our statement. To enhance the clarity of the statement and provide additional evidence, we have added a reference to the corresponding table in the manuscript where the relevant information in presented. We have also included the names of the confirmed NTSR ABC transporter genes to the corresponding phylogenetic tree.

12. Line 551. A depiction of your UGTs that are confirmed NTSRs in the phylogeny would…

Response: We appreciate the reviewer’s suggestion to provide a depiction of the UGTs confirmed as NTSRs in the phylogeny. To address this suggestion, we have added visual representations of the NTSRs within the UGT phylogenetic tree, highlighting their specific position.

Once again, we would like to express our gratitude to the reviewers for their thorough evaluation of our manuscript and their constructive feedback. We believe that the revisions we have made address the concerns raised, resulting in a more accurate and robust study. We hope that the revised manuscript is now suitable for publication in PLOS ONE.

Sincerely,

Samuel Caygill and Liam Dolan.

---

## [Editor Report · Decision Letter 1]

29 Aug 2023

ATP binding cassette transporters and uridine diphosphate glycosyltransferases are ancient protein families that evolved roles in herbicide resistance through exaptation

PONE-D-23-17239R1

Dear Dr. Dolan,

We’re pleased to inform you that your manuscript has been judged scientifically suitable for publication and will be formally accepted for publication once it meets all outstanding technical requirements.

Kind regards,

Yanbin Yin

Academic Editor

PLOS ONE
---

## [Editor Report · Acceptance letter]

13 Sep 2023

PONE-D-23-17239R1 

ATP binding cassette transporters and uridine diphosphate glycosyltransferases are ancient protein families that evolved roles in herbicide resistance through exaptation 

Dear Dr. Dolan:

I'm pleased to inform you that your manuscript has been deemed suitable for publication in PLOS ONE. Congratulations! Your manuscript is now with our production department. 

Kind regards, 

on behalf of

Dr. Yanbin Yin 

Academic Editor

PLOS ONE